# The effect of maternal HMB supplementation on bone mechanical and geometrical properties, as well as histomorphometry and immunolocalization of VEGF, TIMP2, MMP13, BMP2 in the bone and cartilage tissue of the humerus of their newborn piglets

Agnieszka Tomczyk-Warunek[1], Tomasz Blicharski[1]*, Jaromir Jarecki[1], Piotr Dobrowolski[2], Siemowit Muszyński[3], Ewa Tomaszewska[4], Lucio C. Rovati[5]

1 Chair and Department of Rehabilitation and Orthopaedics, Medical University in Lublin, Lublin, Poland, 2 Department of Functional Anatomy and Cytobiology Animal Physiology, Maria Curie-Skłodowska University, Lublin, Poland, 3 Department of Biophysics, University of Life Sciences in Lublin, Lublin, Poland, 4 Department of Animal Physiology, University of Life Sciences in Lublin, Lublin, Poland, 5 School of Medicine and Surgery, University of Milano-Bicocca, Monza, Italy

* blicharski@vp.pl

## Abstract

The presented experiment focuses on assessing the impact of HMB (hydroxy-β-methobuty-rate) supplementation of mothers during pregnancy on the development of the skeletal system of their offspring. For this purpose, an experiment was carried out on 12 clinically healthy sows of the Great White Poland breed, which were divided randomly into two groups the control and the HMB group. All animals were kept under standard conditions and received the same feed for pregnant females. In contrast, females from the HMB group between 70 and 90 days were supplemented with 3-hydroxy-3-methylbutyle in the amount of 0.2g/kg b.w/day. Immediately after birth, the piglets were also divided into groups based on: sex, and presence or lack HMB supplementation, and subsequently were euthanized and humerus bones from all piglets were collected. Mother's HMB supplementation during pregnancy affected the multiple index of their offspring. The higher humerus mass and length was observed with the greater effect in males. Maternal supplementation also influenced on the geometrical and mechanical properties of the humerus as in the case of mass, this effect was higher in males. Also, the collagen structure of the compacted and trabecular bone changed under the HMB addition. Maternal supplementation also affected the expression of selected proteins in growth cartilage and trabecular bone. The obtained results show that the administration to the mother during pregnancy by the HMB significantly affects the development of the humerus in many ways. The obtained results also confirm the utility of such experiments in understanding of the importance of the pregnancy diet as an develop and adaptable factor of offspring organisms and are the base for further research in that area as well as in the protein markers expression area.

**Data Availability Statement:** All relevant data are within the manuscript.

**Funding:** The authors received no specific funding for this work.

**Competing interests:** NO authors have competing interests.

# Introduction

The growth and ossification of individual skeletal elements during prenatal development is different. Studies in animal models showed that the humerus growth and ossification begins much faster than the other long bones, which means that it reaches its length faster [1]. Differences in the development of individual skeletal bones may contribute to the appearance of different adaptive changes during fetal development as a result of environmental changes [2]. Studies in animal models (including pigs whose genome is much more similar to the human genome compared to other animal models) show that undernutrition of the mother during pregnancy significantly affects the skeletal system of her offspring. The effect of maternal nutrition during pregnancy on fetal development has also been confirmed in human studies [2]. The pigs compared to other animal models (mice, rats) are characterized by a very similar size and physiology of internal organs to humans, therefore it is a very good model in preclinical studies and in nutritional experiments [3].

Changes in the skeletal system during prenatal development may in later life contribute to the earlier appearance of osteopenia, as a consequence, osteoporosis [4–7]. As a result of osteoporotic changes, fractures occur not only in the proximal femur, but also in the proximal humerus or distal radius [8].

One of the nutritional factors studied is the supplementation of hydroxy-β-methobutyrate (HMB), which occurs naturally in the body. It is mainly responsible for the proper functioning of the cell membrane, and also participates in *de novo* cholesterol synthesis [9]. It is formed as a result of the degradation of the branched-chain amino acid leucine in the body. Only 5% of the leucine is used for the production of hydroxy-β-methobutyrate [10, 11]. In human nutrition, HMB is used as a supplement for athletes, which is associated with a reduction of the body fat, and an increase in the physical strength of the muscle tissue. This agent has also an anti-catabolic activity, inhibiting protein degradation and protecting muscles against damage [9, 12]. This proves that HMB supplementation can be used as a therapeutic agent in the treatment of elderly sarcopenia. This supplement is also used during the convalescence period [13]. It is not recommended during pregnancy. However, there is no information about any negative effects of using this supplement on fetal development during this period. That is why this supplement has become an area of research interest in recent years. There are few experimental studies using animal models explaining the effect of the HMB supplementation in the prenatal period. However, in the last few years some have been related to the influence of this supplement on the skeletal system [14–20]. In the literature the information on the impact of this supplement on the femoral bone of newborn organisms whose mothers had the HMB supplementation during pregnancy [16, 17]. In these studies, the sows were supplemented between days 70 and 90 of pregnancy, and HMB was administered at a daily dose of 0.2 g/kg of body weight. However, there is no information on the impact of HMB maternal supplementation on the development of the humerus of their newborn offspring. Other ontogenetic development as well as different functions of the humerus may indicate that, in addition to the femur, it may be a model bone for studying skeletal system [1, 5, 21]. The humerus is also an object of interest for researchers, because in adulthood fractures of the proximal humerus occur more frequently than fractures of the proximal femur [22].

An indicator of changes in the skeletal system of offspring of mothers supplemented with HMB may not only be the basic bone properties such as: length, mass, mechanical, geometric properties, histomorphometry of bone and cartilage tissue. Also protein expression and immunolocalization in bone and cartilage can be used as an indicator [17]. For this purpose: VEGF (Vascular endothelial growth factor), TIMP2 (Tissue inhibitor of metalloproteinases), MMP13 (Matrix metalloproteinase 13), BMP2 (Bone morphogenetic protein 2) expressions can be

determined. VEGF is a signal protein that stimulates cell function and is responsible for the formation of blood vessels. As is known, adequate blood supply to bone promotes metabolism [23]. BMP2 stimulates the development of bone and cartilage tissue by influencing the formation of type II collagen fibers [24, 25]. MMP13 or collagenase-3 is responsible for the resorption of bone and cartilage, by degradation of collagen fibers mainly type II but also type I found in the matrix of these tissues. It shows strong expression in the course of embryonic development and can influence the process of modeling these tissues. TIMP2 is an MMP13 antagonist. The available literature also reports that BMP 2 also inhibits the action of MMP13 [26]. The interaction between of the above proteins determines the proper development of the bone [27].

To our knowledge, there are also no studies that show the effect of maternal HMB supplementation on the immunolocalization of VEGF, BMP2, TIMP2, MMP13 which significantly affect the osteolytic and osteogenic processes in the bone and growth plate cartilage in the offspring. There is also no information whether maternal supplementation influences the prenatal development of the humerus, although the literature reports that it affects the growing of the femur. As it known that these two bones are characterized by different ontogenetic development, in order to complete the knowledge the aim of the study was to examine the effect of HMB supplementation on femora development in the pig model offspring.

## Material and methods

### Experimental model

The study was conducted in accordance with the requirements for the use of laboratory animals, which were included in the regulation of the Minister of Agriculture and Rural Development and the Minister of Science and Higher Education. The experiment was accepted by the Local Ethical Committee of Animals at the University of Life Sciences in Lublin (Resolution No. 30/2013 of 16/04/2013). All stages of the experiment were carried out in accordance with its requirements. While conducting the experiment, every effort was made to minimize the suffering and stress of animals.

The experiment was conducted on 24 newborn piglets "before first colostrum uptake" born by 12 sows of Large White Polish All experimental animals were clinically healthy and bred to the same boar. Primiparous sows were used in the study to avoid the impact of the number of pregnancies on birth weight in the offspring. The females used in this experiment were of similar age—10 mouths and weight—between 100 to 120 kg. Animals were kept in separated cages, with the temperature and the humidity control on a 12-hour daily cycle. The females were had free access to fresh water. The pregnant sows were fed twice a day (2,3kg), with balanced standard compound feed for pregnant and lactating sows [16]. The pregnant females randomly divided in to two groups: control (n = 6) and experimental (HMB, n = 6). The experimental group was supplemented, with HMB of the dose 0.2g HMB/kg of the body weight/day between 70 and 90 days of the pregnancy during morning fed. For supplementation was used HMB in the form of calcium salts, while pregnant sows from control groups were supplementation with placebo—$CaCO_3$ at the same dose. This HMB dose was selected based on previous studies that used the same dose of hydroxy-β-methylbutyrate and had a significant effect on offspring development [16, 17, 28]. In this study we decided to supplement sows between 70 to 90 days of pregnancy, because this is critical moment to prenatal development in piglets [29]. During the experiment, no painkillers were used because the experiment was based on supplementation of pregnant mothers by adding HMB or a placebo to wholesome feed.

All piglets born by physiological partum. No difference in body weight of newborn piglets was observed (Grubbs statistic), therefore 6 individuals from each sex were randomly selected

from both groups (one female and one male from one mother). Next newborn piglets were subjected to euthanasia using lethal doses of 0,6 ml/kg of body weight, pentobarbitalum natricum (Morbital, Biowet, Puławy, Poland). Before euthanasia, piglets were premedicated with xylazine at a dose of 1.0 ml / kg of body weight (Sedazin, Biowet, Puławy, Poland). Both humerus were isolated and cleaned of soft tissue. Bones were weighed and measured, an then frozen of -25˚C for s for storage until further analysis. The right humerus were intended for geometric measurements and tissue density analyzes. The humerus taken from left side of the body were intended for mechanical analyzes and histological examination.

## Bone mechanical properties

Before proceeding to the test, all bones of the humerus were thawed at the room temperature. The mechanical properties were examined using a three-point bending test. The test was carried out using a Zwick Z010 testing machine (Zwick GmbH & Company KG, Ulm, Germany), with at constant load [30]. The bones were placed on supports. The distance between supports was 40% of the total bone length. On the basis of the recorded force-deflection curves, the maximum elastic strength and the ultimate strength were determined [17, 30].

## Tissue density of the bone and the cartilage

The study was carried out in the proximal humerus for both bone and cartilage. Prior to the measurement, the test material was dried at 105˚C to remove water. The study was carried out with a Micrometrics AccuPyc 1330 helium pycnometer (Micromeritics, Inc., Norcross, GA) The results of BTD were expressed in g/cm$^3$ [30].

## Bone geometric properties

Right humerus were cut across in the middle of the shaft using a diamond bandsaw 140 (MBS 240/E, Proxxon GmbH, Foehren, Germany). Then the diameters: external horizontal (H) and internal (h), vertical external (B) and internal (b) were measured with a digital caliper. Based on which were calculated: the cortical cross-sectional area (A), the mean relative wall thickness (MRWT) and the cortical index (Cl) were calculated. In addition, a secondary moment of the inertia (Ix) was calculated [30]. The study of geometric features describes the following formulas:

- A—cross-sectional area:

$$A = \frac{\pi}{4}(H \cdot G - h \cdot b)$$

- MRWT—mean relative wall thickness:

$$MRWT = \frac{\left[\left(\frac{B-b}{B}\right) + \left(\frac{H-h}{H}\right)\right]}{2}$$

- Cl—cortical index:

$$Cl = \frac{\left[\left(\frac{B-b}{b}\right) + \left(\frac{H-h}{h}\right)\right]}{2} \cdot 100\%$$

- Ix—second moment of inertia about horizontal axis:

$$l \quad = \frac{\pi}{64} \left( H \cdot B^3 - h \cdot b^3 \right)$$

## Bone histology

After mechanical properties measurements the proximal end of right humerus was cut. The obtained material was initially fixed in a 4% buffered formalin solution. Then material was decalcified using a buffered 10% EDTA solution. Next, the material was dehydrated in increasing gradient of the EtOH. The bone fragments were prepared for standard histological and microscopic procedure. The material was cut into 4 μm thick sections using the HM360 microtome (Microm, Wolldar, Germany) [31, 32]. For the trabecular bone, the growth plate and the joint cartilage, the Goldner's trichrome staining was performed [33].

PSR (Picrosirus red) staining was performed to analyze the proportion of the mature and immature collagen fibers in the trabecular bone and the cortical bone [18, 34]. Stained sections were analyzed using microscope (Olympus BX63; Olympus, Tokyo, Japan) under normal (Goldner) and polarized light (PSR).

## Histological analysis of the trabecular bone, the growth plate, the articular cartilage and the collagen structure

In the growth plate cartilage the thickness of four zones and total thickness was measured. The first (I) resting zone has a large amount of matrix that separates the individual chondrocytes or isogenic groups. In the second (II) proliferating zone, chondrocytes form columns, their shapes flatten, and they are subject to intense division. In the third (III) hypertrophic zone, cartilage cells increase their volume, lose nuclei, and create regular columns. In the fourth (IV) calcium zone, cartilage cells undergo apoptosis. This is the place where the cartilage goes into the bone [31, 33].

In the articular cartilage, the thickness of three zones and the total thickness were measured. In the first (I) superficial zone, the cartilage cells are small, flattened, and parallel to the surface of the cartilage. The second (II) transition zone has a large number of matrices in which chondrocytes are larger, have a round shape, and are arranged alone or form isogenic groups. In the third (III) deep zone, the number of chondrocytes increases, and they regularly form columns [35, 36]. Obtained images were analyzed using the Olympus cell-Sens Version 1.5 graphics analysis software (Olympus, Tokyo, Japan).

The bone volume (BV) and the bone tissue (TV) based on the number of pixels for the epiphysis and the metaphysis were determined. On the basis of the above-mentioned parameters, the ratio of the bone volume to the bone tissue (BV/TV), which was expressed as a percentage, was determined. In the trabecular bone, the number of trabeculae (Tb. N), the thickness of individual trabeculae (Tb.Th), the distance between individual trabecular (Tb.Sp) were determined [35, 36]. The microscope pictures were analysis using the ImageJ software (Wayne Rasband, National Institute of Mental Health, 165 Bethesda, MD, USA). This program was also used for the analysis of the collagen structure.

## Immunohistochemical analysis

For immunohistochemical analysis, deparaffinization and hydration of sections of bone and cartilage tissue were assigned. The next step was to recover the antigenicity of the tested material by enzymatic reactions, by using the proteinase K (Sigma-Aldrich, St. Louis, MO, USA). This step lasted 10 min and was carried out at 37˚C. Then, the a 3% hydrogen peroxide

solution was used to block the peroxidase activity. The incubation was carried out at the room temperature for 5 min. When this process was completed, the normal serum was applied to slides to block all non-specific reactions. This process lasted 30 min and was carried out at 37°C. The next step was the application of the primary antibody: MMP13 (Abcam, Cambridge, UK, concentration 10–20µg/ml), TIMP2 (Abcam, Cambridge, UK, dilution 1:100), VEGF (Biorbyt, Wuhan, Chine, concentration 0,5mg/ml) and BMP2 (Abcam, Cambridge, UK, dilution 1:250). Incubation lasted all night at 4°C. The next day, a secondary antibody: the biotinylated anti-rabbit immunoglobulin (nr. ab6730, Abcam, Cambridge, UK, dilution 1:200) was applied and incubated for 30 min at 37°C. For negative control for each antibody the same procedure was carried out, but without the first antibody. Next DAB (3,3'-diaminobenzidine tetrahydrochloride DAB, Sigma-Aldrich, St. Louis, MO, USA) was applied and incubated for 15 min at the room temperature. For contras staining was used haematoxylin (Sigma-Aldrich, St. Louis, MO, USA) [17].

Microscopic observations allowed to evaluate immunolocalization of MMP13, TIMP2, VEGF, BMP2 in tubercular bone and growth place cartilage, as negative control (nuclei are blue) or positive control (nuclei are brown). The brown color indicates the presence of proteins [16, 17].

To evaluate the expression of the tested proteins for both the bone trabecula and the growth plate cartilage, ten fields of observation were selected in which all cells as well as cells with a positive response were counted. Then, the proportion of cells with a positive immunohistochemical reaction for MMP13, TIMP2, VEGF, BMP2 was calculated. The results were expressed as a percentage [37].

The intensity of protein expression in the matrix of the growth plate cartilage and bone trabeculae was determined on the basis of gray analysis, which is based on the brightness of pixels. For this purpose, microscopic phots were converted to 8-bit grayscale. The higher the pixel value, the weaker the expression of the tested protein. The analysis was performed using the ImageJ software [38].

## Statistical analysis

The statistical analysis of the obtained results was carried out using the software Statistica 13.1 (TIBCO Software Inc., Palo Alto, CA, USA). All results are presented as mean values with standard deviations (mean ± S.D.). Differences between the results were tested using a two-way ANOVA analysis of variance (sex, supplementation) and Tukey's post-hoc test. The normal distribution of data was checked using the Shapiro-Wilk W test, while the homogeneity of variance was tested using the Levene's test. Statistically significant differences were assumed for P <0.05.

## Results

### Bone basis and geometrical properties

In offspring whose mothers received HMB during pregnancy, the bone weight was significantly higher compared to the control. This effect was observed in both sexes. In males, the use of HMB in their mothers also caused significant increased the length of the humerus. Also, for males there was a significant increase in the vertical and horizontal external diameter, the cross sectional area, the mean wall thickness and the cortical index in comparison to the control group. These effects were not observed in females whose pregnant mothers were supplemented with HMB. In both sexes of the experimental groups a significant increase in the moment of inertia was observed (Table 1).

**Table 1. The effect of maternal treatment with HMB on the humerus osteometric and geometrical properties in newborn piglets.**

| | Osteometric properties | | Geometrical properties | | | | | | | |
|---|---|---|---|---|---|---|---|---|---|---|
| | Weight, g | Lenght, mm | B, mm | b, mm | H, mm | h, mm | A, mm$^2$ | MRWT, — | CI, % | Ix, mm$^4$ |
| **Main effect supplementation** | | | | | | | | | | |
| **Control** | 5.09 | 46.34 | 6.16 | 2.77 | 4.77 | 1.94 | 19.01 | 1.36 | 57.23 | 55.15 |
| **HMB** | 8.50 | 55.00 | 7.09 | 2.70 | 5.49 | 2.22 | 25.99 | 1.63 | 60.74 | 90.97 |
| **Main effect sex** | | | | | | | | | | |
| **Male** | 7.15 | 49.62 | 6.60 | 2.63 | 5.10 | 2.09 | 22.41 | 1.51 | 59.43 | 71.05 |
| **Female** | 6.44 | 51.72 | 6.65 | 2.85 | 5.16 | 2.07 | 22.59 | 1.48 | 58.53 | 75.07 |
| **Treatment effect** | | | | | | | | | | |
| **Control male** | 5.03[a] ± 0.469 | 43.79[b] ± 5.34 | 5.93[a] ± 0.623 | 2.68[a] ± 0.387 | 4.68[a] ± 0.319 | 2.04[a] ± 0.298 | 17.76[a] ± 2.67 | 1.28[a] ± 0.231 | 55.97[a] ± 4.17 | 49.19[a] ± 11.12 |
| **HMB male** | 9.26[b] ± 0.943 | 55.46[a] ± 5.87 | 7.21[c] ± 0.394 | 2.58[a] ± 0.412 | 5.53[c] ± 0.201 | 2.13[a] ± 0.321 | 27.05[c] ± 2.39 | 1.74[c] ± 0.189 | 62.89[b] ± 2.68 | 92.91[b] ± 10.91 |
| **Control female** | 5.14[a] ± 0.598 | 48.90[ab] ± 6.79 | 6.34[ab] ± 0.549 | 2.87[a] ± 0.189 | 4.86[ab] ± 0.534 | 1.83[a] ± 0.289 | 20.26[ab] ± 3.91 | 1.45[ab] ± 0.197 | 58.97[a] ± 3.78 | 61.10[a] ± 14.01 |
| **HMB female** | 7.74[b] ± 0.991 | 54.56[a] ± 5.79 | 6.96[bc] ± 0.578 | 2.83[a] ± 0.361 | 5.46[bc] ± 0.413 | 2.32[a] ± 0.496 | 24.92[bc] ± 3.43 | 1.52[bc] ± 0.159 | 58.58[a] ± 4.02 | 89.03[b] ± 9.88 |
| **Pooled SEM** | 0.53 | 1.67 | 0.20 | 0.17 | 0.15 | 0.14 | 1.2 | 0.06 | 0.87 | 6.27 |
| **Main effect and interaction** | | | | | | | | | | |
| **Supplementation** | <0.001 | <0.001 | <0.001 | 0.701 | <0.001 | 0.066 | <0.001 | <0.001 | <0.001 | <0.001 |
| **Sex** | 0.261 | 0.234 | 0.793 | 0.220 | 0.728 | 0.938 | 0.885 | 0.688 | 0.322 | 0.566 |
| **Supplementation x sex** | 0.195 | 0.093 | 0.147 | 0.851 | 0.432 | 0.198 | 0.076 | 0.004 | 0.010 | 0.264 |

[a, b, c]—mean values in rows with different letters differ significantly at P<0.05; SEM—standard error of the means.

H—horizontal external diameter; h—horizontal internal diameter; B—vertical external diameter; b—vertical internal diameter; A—cortical cross-sectional area; MRWT—the mean relative wall thickness; Cl—the cortical index; Ix—secondary moment of inertia.

## Bone mechanical properties

The use of the HMB in the maternal nutrition contributed to a significant increase in ultimate strength and maximal elastic strength in their offspring compared to newborns in the control group. This effect was observed in both sexes (Table 2).

## Histomorphometry of the proximal humerus epiphysis, metahpysis and BTD

Histomorphometric analysis of the epiphysis revealed a significantly higher number of trabeculae in females compared to males. Also these differences were observed after HMB treatment. In the case of the relative bone volumes as well as the trabecular space, histomorphometric analysis revealed a significantly lower value of these parameters in females as compared to males. However, in the case of BV/TV in females, supplementation of their mothers during pregnancy contributed to the increase of this parameter. This effect was not observed in males. The size of the trabecular space did not change under the influence of HMB treatment. The trabecular thickness was significantly higher in the males control group compared to males from the experimental group. This effect was not observed in females. Sex also did not affect this parameter (Table 3).

The males from the control group were characterized by higher the BTD than females from the same group. In males there was a significant decrease in the BTD from the HMB group compared to the control group, this effect was not observed in females (Table 3).

**Table 2. The effect of maternal treatment with HMB on mechanical properties in newborn piglets.**

| | Ultimate strength, N | Max. elastic strength, N |
|---|---|---|
| **Main effect supplementation** | | |
| **Control** | 189.17 | 89.58 |
| **HMB** | 299.42 | 178.33 |
| **Main effect sex** | | |
| **Male** | 226.33 | 138.33 |
| **Female** | 262.25 | 129.58 |
| **Treatment effect** | | |
| **Control male** | 167.33[a] ± 32.03 | 88.33[a] ± 10.61 |
| **HMB male** | 285.33[bc] ± 49.94 | 188.33[b] ± 18.91 |
| **Control female** | 211.00[ab] ± 45.33 | 90.83[a] ± 17.17 |
| **HMB female** | 313.50[c] ± 32.25 | 168.33[b] ± 19.24 |
| **Pooled SEM** | 18.92 | 11.49 |
| **Main effects and interaction** | | |
| **Supplementation** | <0.001 | <0.001 |
| **Sex** | 0.079 | 0.515 |
| **Supplementation x sex** | 0.694 | 0.404 |

[a, b, c]—mean values in rows with different letters differ significantly at P<0.05; SEM—standard error of the means.

**Table 3. The effect of maternal treatment with HMB on humerus epiphysis and metaphysis histomorphology in newborn piglets.**

| | Epiphysis | | | | | Metaphysis | | | |
|---|---|---|---|---|---|---|---|---|---|
| | Tb. N/mm of bone | BV/TV, % | Tb.Th mean, μm | Tb.Sp mean, μm | BTD, g/cm³ | Tb. N/mm of bone | BV/TV, % | Tb.Th mean, μm | Tb.Sp mean, μm |
| **Main effect supplementation** | | | | | | | | | |
| **Control** | 8.29 | 12.48 | 26.53 | 123.68 | 2.48 | 4.69 | 11.53 | 22.83 | 261.04 |
| **HMB** | 8.58 | 21.84 | 25.40 | 119.38 | 2.22 | 4.65 | 10.64 | 21.89 | 285.35 |
| **Main effect sex** | | | | | | | | | |
| **Male** | 7.05 | 15.73 | 25.99 | 142.86 | 2.42 | 4.22 | 9.36 | 21.90 | 292.21 |
| **Female** | 9.82 | 15.60 | 25.91 | 100.21 | 2.29 | 5.13 | 12.81 | 22.82 | 254.18 |
| **Treatment effect** | | | | | | | | | |
| **Control male** | 6.85[a] ± 1.03 | 19.10[a] ± 3.00 | 27.15[b] ± 2.48 | 143.46[b] ± 19.62 | 2.61[b] ± 0.232 | 3.79[a] ± 0.931 | 8.42[a] ± 2.50 | 22.04[a] ± 2.74 | 303.79[a] ± 91.01 |
| **HMB male** | 7.25[a] ± 1.23 | 18.01[a] ± 3.01 | 24.83[a] ± 1.65 | 142.27[b] ± 24.04 | 2.23[a] ± 0.071 | 4.64[ab] ± 1.01 | 10.30[ab] ± 3.99 | 21.76[a] ± 2.91 | 218.28[a] ± 91.04 |
| **Control female** | 9.72[b] ± 1.22 | 15.77[b] ± 2.01 | 25.91[ab] ± 3.18 | 103.92[a] ± 15.67 | 2.36[a] ± 0.113 | 5.59[b] ± 0.892 | 14.64[b] ± 2.99 | 23.63[a] ± 2.99 | 280.62[a] ± 63.09 |
| **HMB female** | 9.91[b] ± 0.741 | 25.68[c] ± 1.99 | 25.90[ab] ± 1.89 | 96.51[a] ± 11.24 | 2.22[a] ± 0.067 | 4.66[ab] ± 1.25 | 10.99[ab] ± 3.67 | 22.01[a] ± 3.22 | 290.07[a] ± 78.01 |
| **Pooled SEM** | 1.23 | 3.01 | 1.98 | 27.85 | 0.12 | 1.72 | 4.67 | 3.55 | 125.51 |
| **Main effects and interaction** | | | | | | | | | |
| **Supplementation** | 0.366 | <0.001 | 0.029 | 0.571 | <0.001 | 0.930 | 0.471 | 0.317 | 0.454 |
| **Sex** | <0.001 | <0.001 | 0.876 | <0.001 | 0.032 | 0.048 | 0.006 | 0.332 | 0.243 |
| **Supplementation x sex** | 0.450 | <0.001 | 0.331 | 0.682 | 0.038 | 0.053 | 0.027 | 0.481 | 0.146 |

[a, b, c]—mean values in rows with different letters differ significantly at P<0.05; SEM—standard error of the means.

BV/TV—relative bone volume; Tb.Th—trabecular thickness; Tb.Sp—trabecular separation; Tb.N—trabecular number; BTD—bone tissue density.

Histomorphometric analysis showed that HMB did not affect histomorphometry of the bones in the metaphysis. However, it was observed that females in the control group were characterized by the significantly larger number and volume of trabeculae compared to males in the control group (Table 3).

## Distribution of immature and mature collagen fibers

In the cortical bone the HMB supplementation of mothers during pregnancy significantly increased the share immature collagen fibers in females. In contrast, in males, maternal supplementation contributed to a decrease in mature collagen fibers. Also, a significant decrease in the ratio of mature to immature fiber collagen was also observed in males (Table 4, Fig 1).

In the epiphysis of the humerus in males from the HMB group, a significant decrease in the share of immature and mature collagen fibers as well as a decrease in their ratio was observed. An increase in the share of immature collagen fibers and a decrease in the ratio of mature to immature collagen in females from the HMB group was observed (Table 4, Fig 1).

In the methaphysis in females from the HMB group compared to the control group, an increase in the share of immature collagen fibers and the mature collagen fibers was noted. In males, supplementation with their mothers contributed to a reduction in the content of immature collagen fibers and an increase in the ratio of mature to immature collagen (Table 4, Fig 1).

**Table 4. The effect of maternal treatment with HMB on humerus collagen structure cortical and tubercular bone in newborn piglets.**

| | Tubercular bone | | | | | | Cortical bone | | |
|---|---|---|---|---|---|---|---|---|---|
| | Epiphysis | | | Metaphysis | | | | | |
| | Immature collagen fibers, % | Mature collagen fibers, % | The ration of mature to immature collagen, — | Immature collagen fibers, % | Mature collagen fibers, % | The ration of mature to immature collagen, — | Immature collagen fibers, % | Mature collagen fibers, % | The ration of mature to immature collagen, — |
| **Main effect supplementation** | | | | | | | | | |
| **Control** | 2.25 | 20.16 | 8.59 | 1.29 | 5.68 | 4.37 | 2.56 | 31.14 | 11.74 |
| **HMB** | 2.22 | 17.53 | 6.03 | 0.76 | 5.80 | 4.03 | 3.37 | 27.52 | 8.74 |
| **Main effect sex** | | | | | | | | | |
| **Male** | 2.11 | 18.19 | 7.53 | 1.15 | 7.29 | 14.79 | 2.75 | 25.11 | 9.24 |
| **Female** | 2.26 | 19.50 | 7.09 | 1.27 | 7.61 | 6.45 | 3.29 | 33.56 | 11.17 |
| **Treatment effect** | | | | | | | | | |
| **Control male** | 2.50[bc] ± 0.612 | 22.03[a] ± 5.77 | 8.88[b] ± 2.56 | 2.04[d] ± 0.443 | 7.26[ab] ± 1.51 | 4.36[a] ± 2.53 | 2.41[a] ± 0.521 | 29.52[a] ± 4.78 | 11.22[a] ± 2.11 |
| **HMB male** | 1.72[a] ± 0.821 | 14.35[b] ± 4.33 | 6.18[a] ± 2.60 | 0.26[a] ± 0.012 | 7.32[ab] ± 0.991 | 25.22[b] ± 3.77 | 2.09[ab] ± 0.632 | 20.70[b] ± 4.82 | 7.26[b] ± 3.07 |
| **Control female** | 1.99[ab] ± 0.608 | 18.28[ab] ± 5.03 | 8.30[b] ± 2.06 | 1.01[b] ± 0.290 | 6.35[a] ± 1.21 | 6.74[a] ± 2.56 | 2.72[a] ± 0.910 | 32.77[a] ± 4.02 | 12.14[a] ± 3.21 |
| **HMB female** | 2.73[c] ± 0.709 | 20.72[a] ± 3.89 | 5.87[a] ± 2.71 | 1.52[c] ± 0.210 | 8.87[b] ± 1.01 | 6.17[a] ± 2.31 | 3.86[b] ± 1.81 | 34.35[a] ± 4.88 | 10.21[ab] ± 2.66 |
| **Pooled SEM** | 0.871 | 6.67 | 2.98 | 0.820 | 1.32 | 1.13 | 1.65 | 8.54 | 3.98 |
| **Main effects and interaction** | | | | | | | | | |
| **Supplementation** | 0.864 | 0.019 | <0.0001 | <0.0001 | 0.055 | <0.0001 | <0.001 | 0.023 | <0.001 |
| **Sex** | 0.080 | 0.239 | 0.439 | 0.818 | 0.635 | <0.0001 | 0.039 | <0.000 | 0.024 |
| **Supplementation x sex** | <0.0001 | <0.0001 | 0.812 | <0.0001 | 0.066 | <0.0001 | 0.382 | <0.001 | 0.235 |

[a, b, c]—mean values in rows with different letters differ significantly at P<0.05; SEM—standard error of the means.

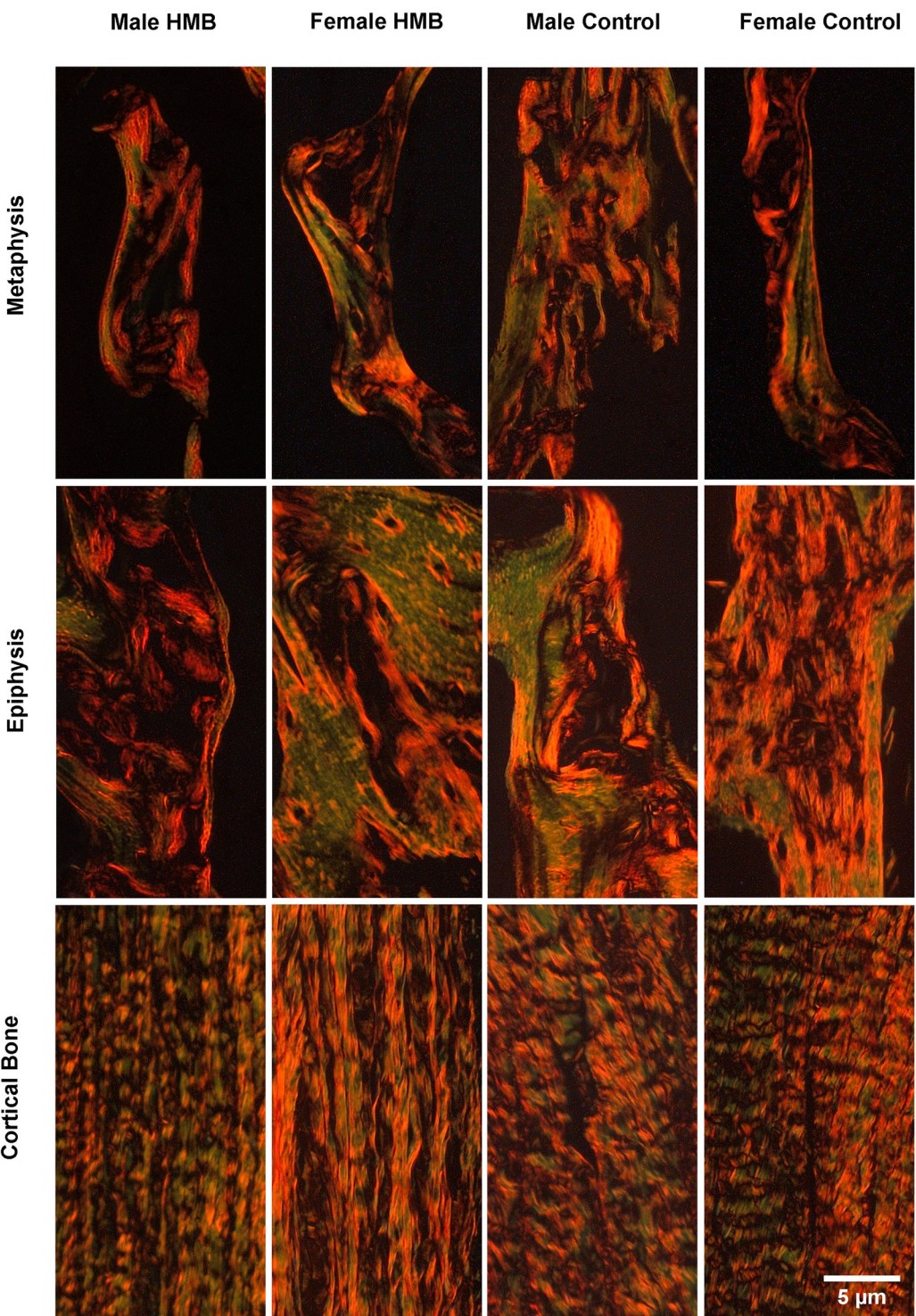

**Fig 1. Representative images of PSR (Picrosirus red) staining carried out on formaldehyde-fixed fragments sections of the trabecular bone of the humerus metaphysis, epiphysis and the cortical bone of newborn piglets.** Mature collagen fibers are red or orange, while immature collagen fibers are green.

**Table 5. The effect of maternal treatment with HMB on humerus growth plate cartilage histomorphology in newborn piglets.**

| | Zone I | Zone II | Zone III | Zone IV | Total thickness |
|---|---|---|---|---|---|
| **Main effect supplementation** | | | | | |
| **Control** | 208.14 | 191.08 | 153.13 | 191.66 | 879.15 |
| **HMB** | 208.18 | 238.56 | 203.17 | 185.84 | 918.18 |
| **Main effect sex** | | | | | |
| **Male** | 191.55 | 223.16 | 175.82 | 208.07 | 918.41 |
| **Female** | 225.90 | 206.48 | 180.49 | 169.43 | 878.91 |
| **Treatment effect** | | | | | |
| **Control male** | 218.49$^c$ ± 28.01 | 200.18$^c$ ± 24.08 | 153.31$^a$ ± 24.78 | 207.95$^b$ ± 27.15 | 942.78$^a$ ± 69.17 |
| **HMB male** | 164.61$^a$ ± 22.83 | 246.78$^a$ ± 27.01 | 200.32$^b$ ± 28.59 | 208.16$^b$ ± 27.31 | 894.05$^a$ ± 90.47 |
| **Control female** | 199.87$^b$ ± 21.91 | 181.97$^b$ ± 19.81 | 152.96$^a$ ± 27.18 | 175.37$^a$ ± 28.78 | 815.52$^b$ ± 72.04 |
| **HMB female** | 251.67$^d$ ± 29.24 | 230.34$^a$ ± 32.24 | 208.02$^b$ ± 20.11 | 163.49$^a$ ± 20.74 | 942.30$^a$ ± 103.59 |
| **Pooled SEM** | 4.38 | 7.14 | 5.15 | 8.09 | 27.32 |
| **Main effects and interaction** | | | | | |
| **Supplementation** | 0.008 | <0.001 | <0.001 | 0.480 | <0.005 |
| **Sex** | <0.001 | <0.001 | 0.384 | <0.001 | 0.114 |
| **Supplementation x sex** | <0.001 | 0.852 | 0.349 | 0.462 | 0.211 |

[a, b, c, d]—mean values in rows with different letters differ significantly at P<0.05; SEM—standard error of the means.

Zone I—resting zone; Zone II—proliferating zone; Zone III—hypertrophic zone; Zone IV—calcium zone.

## Histomorphometry of the growth plate cartilage

Histomorphometric analysis of growth plate cartilage revealed significantly higher thickness the zone I in male compare to females. The HMB supplementation of females contributed to an increase in the thickness the zone I in comparison to the control group. However, we see the reverse effect in males. In both sexes, the administration of the HMB to the mother during pregnancy led to an increase thickness the zone II, with no differences between the sexes. Although in the control groups males had significantly thicker zone II. The zone III is thicker with both sexes coming from experimental groups. Histomorphometric analysis revealed significantly higher thickness the zone IV in males compared to females and no effect of HMB treatment. There was no significant statistical difference in the total thickness of the growth cartilage between the males in the group, whose mothers were supplemented with the HMB and the thick control. However, a significant increase in this parameter was observed in females from the experimental group compared to the control group (Table 5).

## Histomorphometry of the articular cartilage and BTD

In both females and males of mothers who were the HMB supplemented during pregnancy, the thickness of the I zone of the articular cartilage was increased. This effect can also be observed in males in the II zone of this cartilage, while in females the administration of the HMB reduced the thickness of this zone. The administration of the HMB to the mother during pregnancy contributed to the increase in the thickness of the III zone in males, while no effect was seen in females. The total thickness of the articular cartilage was significantly higher in both sexes of animals from the experimental group as compared to the control group. The effect of the HMB administration during pregnancy was higher in males than in females. Only in the case of the thickness zone I this effect was higher in females. The BTD reduced in the

**Table 6. The effect of maternal treatment with HMB on humeral articular cartilage histomorphology and BTD in newborn piglets.**

| | Zone I | Zone II | Zone III | Total thickness | BTD |
|---|---|---|---|---|---|
| **Main effect supplementation** | | | | | |
| **Control** | 13.95 | 565.96 | 332.78 | 1082.41 | 2.21 |
| **HMB** | 17.88 | 542.65 | 421.67 | 1190.97 | 1.97 |
| **Main effect sex** | | | | | |
| **Male** | 13.97 | 536.71 | 382.03 | 1178.54 | 1.92 |
| **Female** | 17.87 | 571.90 | 372.41 | 1094.84 | 2.01 |
| **Treatment effect** | | | | | |
| **Control male** | $11.60^b \pm 4.83$ | $504.68^a \pm 52.03$ | $300.98^b \pm 46.70$ | $1128.50^a \pm 99.01$ | $2.25^b \pm 0.297$ |
| **HMB male** | $16.31^a \pm 2.28$ | $568.73^b \pm 51.22$ | $463.08^c \pm 58.62$ | $1228.57^c \pm 98.72$ | $1.78^a \pm 0.104$ |
| **Control female** | $16.42^a \pm 2.17$ | $627.24^c \pm 60.81$ | $364.57^a \pm 49.86$ | $1036.31^b \pm 76.42$ | $2.01^{ab} \pm 0.127$ |
| **HMB female** | $19.42^b \pm 3.14$ | $516.57^a \pm 50.72$ | $380.25^a \pm 54.70$ | $1153.37^a \pm 63.22$ | $1.83^a \pm 0.164$ |
| **Pooled SEM** | 0.65 | 13.53 | 8.83 | 16.92 | 0.07 |
| **Main effects and interaction** | | | | | |
| **Supplementation** | <0.001 | 0.094 | <0.001 | <0.001 | <0.001 |
| **Sex** | <0.001 | 0.012 | 0.286 | <0.001 | 0.260 |
| **Supplementation x sex** | 0.208 | <0.001 | 0.081 | 0.679 | 0.063 |

[a, b, c]—mean values in rows with different letters differ significantly at P<0.05; SEM—standard error of the means.

Zone I—superficial; Zone II—transition zone; Zone III—deep zone. BTD—bone tissue density.

HMB males, whereas in females the differences were not observed. Sex did not affected the HMB for this parameter (Table 6).

## Vascular endothelial growth factor (VEGF) immunolocalization in the bone trabecula and the growth plate cartilage

VEGF staining showed a weak response in the bone trabecular matrix and the cytoplasm of cells in both study groups, regardless of gender, compared to the negative control. Statistical analysis in the case of a positive response to the presence of VEGF in the cytoplasm of cells, showed that males from the control group had a greater response compared to males from the HMB group. On the other hand, in females, an inverse relationship was observed (Table 7. Fig 2).

Analysis of the expression of this protein in the bone trabecular matrix based on the gray analysis showed that males from the HMB group were characterized by a smaller response compared to the other groups (Table 8, Fig 2).

In the case of growth plate cartilage, a response in the matrix in both test groups irrespective of sex compared to the negative control was observed. Statistical analysis showed that, regardless of the group, females had a higher proportion of cartilage cells with a positive reaction. In this sex as well, microscopic observation showed a strong staining of the cartilage cells in zones II and III compared to males (many brown-stained nuclei and cells). Supplementation did not significantly affect the percentage of cells with a positive response to the presence of VEGF in their cytoplasm. In control females, strong VEGF staining was also observed in the cell nuclei in the proliferative zone (Table 7, Figs 3 and 4).

In the cartilage tissue matrix, the strongest reaction was observed in zone IV in all groups, especially at the border of this zone with zone III. Gray-scale analysis of the expression of this protein at the border of these zones showed that females from the HMB group had the strongest VEGF expression compared to the other groups (Table 8, Figs 3 and 4).

**Table 7. The effect of maternal treatment with HMB on humerus growth plate cartilage and trabecular bone cells with a positive immunohistochemical reaction for MMP13, TIMP2, VEGF, BMP2 in newborn piglets.**

| | Growth plate cartilage | | | | Trabecular bone | | | |
|---|---|---|---|---|---|---|---|---|
| | MMP13, % | TIMP2, % | VEGF, % | BMP2, % | MMP13, % | TIMP2, % | VEGF, % | BMP2, % |
| **Main effect supplementation** | | | | | | | | |
| **Control** | 42.48 | 73.89 | 73.30 | 69.38 | 90.94 | 92.89 | 73.45 | 90.76 |
| **HMB** | 59.48 | 85.40 | 72.48 | 89.34 | 97.06 | 94.33 | 69.90 | 87.76 |
| **Main effect sex** | | | | | | | | |
| **Male** | 38.71 | 90.60 | 62.94 | 88.56 | 99.98 | 96.89 | 73.53 | 91.37 |
| **Female** | 63.25 | 68.69 | 82.83 | 70.16 | 87.99 | 90.39 | 69.81 | 86.88 |
| **Treatment effect** | | | | | | | | |
| **Control male** | $21.13^b \pm 4.13$ | $94.98^d \pm 5.03$ | $61.49^a \pm 5.77$ | $81.97^a \pm 4.76$ | $91.81^a \pm 9.12$ | $99.00^a \pm 2.24$ | $87.00^b \pm 8.37$ | $93.73^a \pm 5.78$ |
| **HMB male** | $56.28^c \pm 7.01$ | $86.22^c \pm 6.76$ | $64.40^a \pm 9.82$ | $95.15^c \pm 3.87$ | $98.91^a \pm 3.39$ | $94.78^a \pm 5.29$ | $60.07^a \pm 6.55$ | $89.00^a \pm 6.27$ |
| **Control female** | $62.68^a \pm 6.40$ | $75.81^b \pm 8.01$ | $82.19^b \pm 6.12$ | $56.79^b \pm 7.11$ | $81.87^a \pm 6.27$ | $89.78^a \pm 6.27$ | $59.89^a \pm 2.25$ | $85.98^a \pm 3.09$ |
| **HMB female** | $63.83^a \pm 8.56$ | $61.56^a \pm 7.96$ | $83.47^b \pm 9.05$ | $83.53^a \pm 9.79$ | $94.17^a \pm 5.39$ | $91.99^a \pm 5.94$ | $79.72^b \pm 3.60$ | $87.79^a \pm 3.69$ |
| **Pooled SEM** | 3.91 | 4.17 | 1.43 | 1.72 | 1.30 | 1.96 | 2.10 | 2.75 |
| **Main effects and interaction** | | | | | | | | |
| **Supplementation** | <0.001 | <0.001 | 0.053 | <0.001 | 0.521 | 0.552 | 0.090 | 0.556 |
| **Sex** | <0.001 | <0.001 | <0.001 | <0.001 | 0.011 | 0.090 | 0.135 | 0.956 |
| **Supplementation x sex** | <0.001 | <0.001 | <0.001 | <0.001 | 0.522 | 0.579 | 0.665 | 0.020 |

[a, b, c, d]- mean values in rows with different letters differ significantly at P<0.05; SEM—standard error of the means.

MMP13—matrix metalloproteinase 13; TIMP2—tissue inhibitor of metalloproteinases; BMP2—bone morphogenetic protein 2; VEGF—vascular endothelial growth factor.

## Matrix metalloproteinase 13 (MMP13) immunolocalization in the bone trabecula and the growth plate cartilage

The MMP13 staining showed a response in the matrix of trabecular bone in all individuals. Also, in the trabecular bone, the presence of MMP13 in the cell cytoplasm in all groups was observed. In the case of females from the HMB group, statistical analysis showed a significant increase in the proportion of cells with a positive response to the presence of MMP13. Microscopic observation showed that in the HMB group, expression in cells was similar in both sexes. Whereas in the control group, stronger reactions in males was observed. In males, supplementation did not affect the expression of MMP13 in the cytoplasm of cells. However, in females from the HMB group this reaction was stronger than in comparison with females from the control group (Table 7, Fig 2).

Analysis of MMP13 expression in the matrix of trabecular showed a stronger response to the presence of MMP13 in males from HMB group (Table 8).

In the case of growth plate cartilage, a response in the matrix in both test groups irrespective of sex compared to the negative control was observed. Analysis of the proportion of cells with a positive response to the presence of MMP13 cells in the cytoplasm showed that, compared to the control group, males HMB had a stronger response. The same analysis showed that regardless of supplementation, female higher cells responded positively. There was no difference between the females from the control group and the females from the HMB group in this analysis (Table 7). Microscopic observation showed that all groups showed reactions in the cytoplasm of cartilage cells in zones II and III (brown color of the cytoplasm), while there was no reaction in the cell nucleus (blue nucleus) (Figs 3 and 4). In both sexes, supplementation

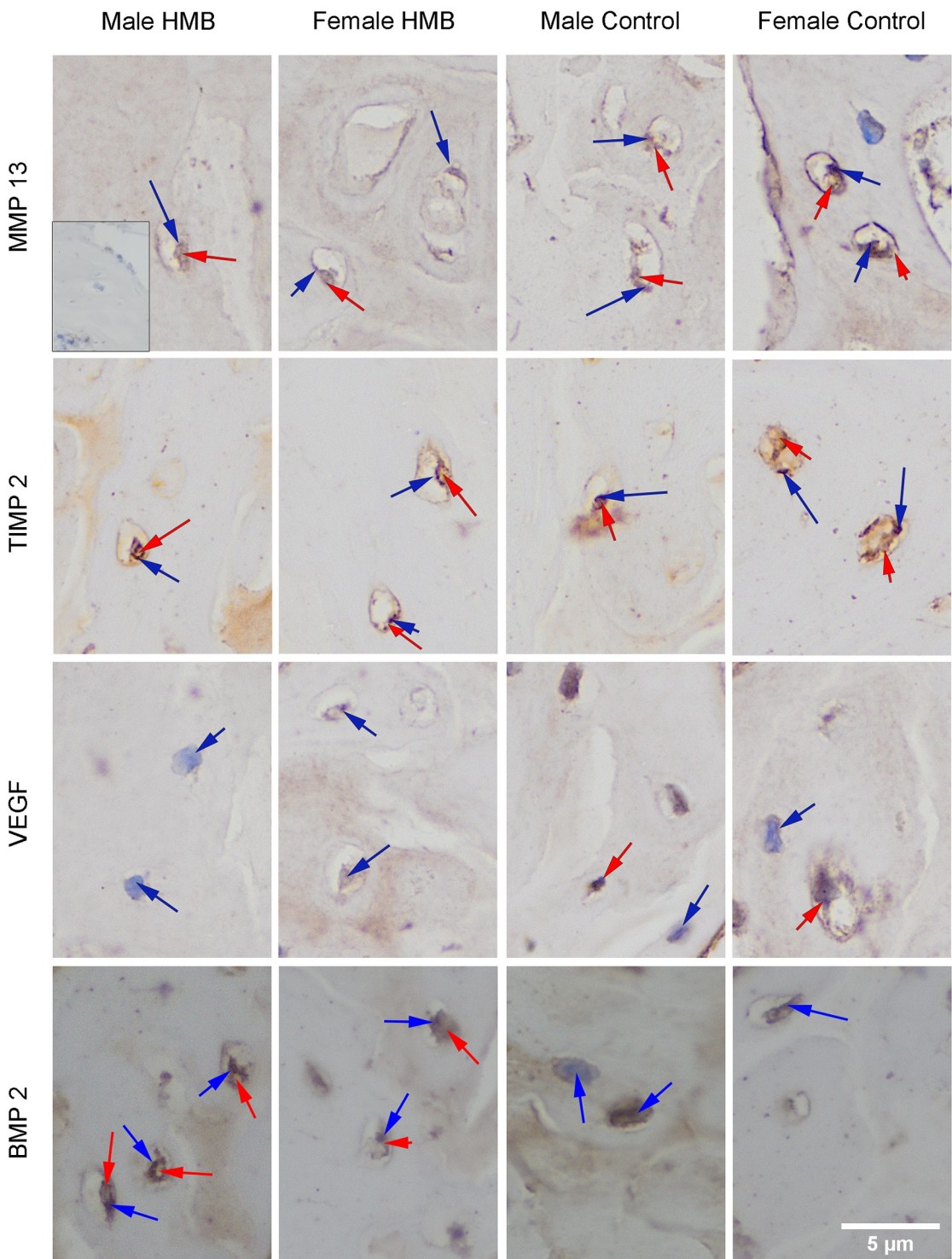

**Fig 2. Imunstohistimechmic analysis of the expression the MMP13 (matrix metalloproteinase 13), the TIMP2 (tissue inhibitor of metalloproteinases 2), the VEFG (vascular endothelial growth factor) and the BMP2 (bone morphogenetic protein 2) from the humerus trabeculae.** Representative images immunohistochemical analysis carried out on the formaldehyde-fixed from the humerus growth cartilage of newborn piglets. The positive reaction (presence of MMP13, TIMP2, VEFG, BMP2) in the cytoplasm or the cell nucleus was marked in red, while the lack of expression was marked in blue.

**Table 8. Effect of maternal HMB treatment on the intensity of immunoreactions against MMP13, TIMP2, VEGF, BMP2 in the matrix between III and IV zone of growth plate cartilage as well as trabecular bone of the humerus in newborn piglets.**

| | Growth plate cartilage | | | | Trabecular bone | | | |
|---|---|---|---|---|---|---|---|---|
| | MMP13, pixel | TIMP2, pixel | VEGF, pixel | BMP2, pixel | MMP13, pixel | TIMP2, pixel | VEGF, pixel | BMP2, pixel |
| **Main effect supplementation** | | | | | | | | |
| **Control** | 167.00 | 126.18 | 165.52 | 174.94 | 176.54 | 181.12 | 178.64 | 137.76 |
| **HMB** | 156.09 | 167.44 | 133.12 | 119.42 | 174.82 | 183.91 | 180.97 | 136.81 |
| **Main effect sex** | | | | | | | | |
| **Male** | 160.40 | 140.92 | 162.13 | 120.75 | 170.76 | 181.25 | 180.48 | 116.33 |
| **Female** | 162.68 | 150.69 | 136.50 | 173.61 | 182.61 | 183.78 | 179.14 | 156.24 |
| **Treatment effect** | | | | | | | | |
| **Control male** | 173.77$^c$ ± 6.80 | 125.52$^a$ ± 6.17 | 165.13$^{ab}$ ± 6.18 | 144.30$^a$ ± 8.66 | 172.98$^a$ ± 6.93 | 178.12$^b$ ± 5.71 | 178.06$^a$ ± 6.57 | 117.05$^a$ ± 6.47 |
| **HMB male** | 147.03$^b$ ± 9.62 | 156.33$^b$ ± 6.01 | 159.13$^a$ ± 7.92 | 97.13$^b$ ± 9.12 | 168.54$^a$ ± 7.60 | 184.38$^a$ ± 7.95 | 182.89$^b$ ± 5.00 | 115.62$^a$ ± 7.81 |
| **Control female** | 160.22$^a$ ± 9.77 | 126.84$^a$ ± 6.05 | 165.90$^b$ ± 7.90 | 205.50$^c$ ± 7.18 | 180.1$^b$ ± 8.01 | 184.12$^a$ ± 7.07 | 179.22$^{ab}$ ± 6.32 | 154.47$^b$ ± 9.56 |
| **HMB female** | 165.14$^a$ ± 5.87 | 174.54$^c$ ± 7.97 | 107.10$^c$ ± 5.48 | 141.71$^a$ ± 4.86 | 185.10$^b$ ± 7.01 | 183.43$^a$ ± 6.23 | 179.05$^{ab}$ ± 3.22 | 158.00$^b$ ± 6.83 |
| **Pooled SEM** | 1.79 | 1.35 | 1.61 | 1.60 | 1.66 | 1.32 | 1.27 | 1.39 |
| **Main effects and interaction** | | | | | | | | |
| **Supplementation** | <0.001 | <0.001 | <0.001 | <0.001 | 0.848 | 0.043 | 0.073 | 0.479 |
| **Sex** | 0.217 | <0.001 | <0.001 | <0.001 | <0.001 | 0.067 | 0.299 | <0.001 |
| **Supplementation x sex** | <0.001 | <0.001 | <0.001 | <0.001 | <0.001 | 0.012 | 0.055 | 0.097 |

$^{a, b, c}$—mean values in rows with different letters differ significantly at P<0.05; SEM—standard error of the means.

MMP13—matrix metalloproteinase 13; TIMP2—tissue inhibitor of metalloproteinases; BMP2—bone morphogenetic protein 2; VEGF—vascular endothelial growth factor; Zone III—hypertrophic zone; Zone IV—calcium zone.

resulted in a stronger expression of this protein in the cytoplasm of cartilage cells (Table 7, Figs 3 and 4).

In the growth plate cartilage matrix the strongest reactions on the border of zones III and IV and in zone IV in all groups were observed. Gray analysis showed the strongest expression of this white in male HMB (Table 8, Figs 3 and 4).

## Tissue inhibitor of metalloproteinases 2 (TIMP2) immunolocalization in the bone trabecula and the growth plate cartilage

The TIMP2 staining showed a response in matrix and bone trabecular cells of all test groups regardless of gender and supplementation, relative to the negative control. The analysis of the participation of cells with a positive reaction to the presence of TIMP2 in the cytoplasm of these cells showed that in the case of males, the control group was characterized by a higher percentage of these cells compared to the HMB group (Table 7, Fig 2).

The analysis of TIMP expression in the trabecular matrix showed a decrease in TIMP expression in males from the HMB group compared to the other groups (Table 8, Fig 2).

In the case of growth plate cartilage, a response in the matrix and cell in all studied groups was observed. The analysis of the proportion of cells with a positive response to the presence of cells in the cytoplasm was significantly higher in both control groups compared to the groups whose mothers were supplemented with HMB. Microscopic observation showed a stronger response in the chondrocyte cell membrane in the groups in which mothers did not receive supplementation. In females from the HMB group, the expression of TIMP2 in the cytoplasm of chondrocytes was weaker compared to the control group. Females from both groups did not express the tested protein in many cell nuclei. However, when comparing the two study

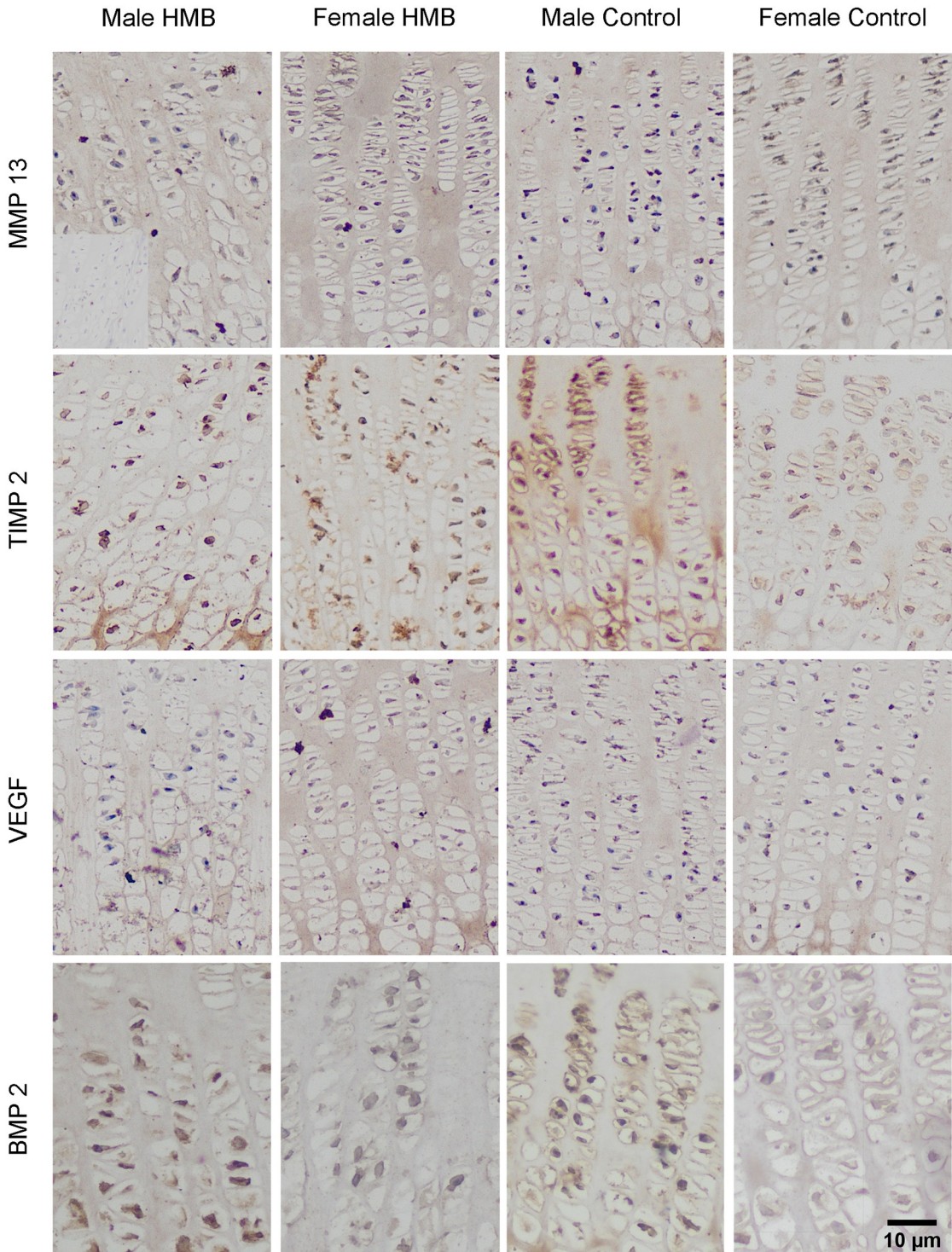

**Fig 3. Immunohistochemical analysis of the immunolocalization the MMP13 (matrix metalloproteinase 13), the TIMP2 (tissue inhibitor of metalloproteinases 2), the VEFG (vascular endothelial growth factor) and the BMP2 (bone morphogenetic protein 2) from the humerus growth plate cartilage.** Representative images immunohistochemical analysis carried out on the formaldehyde-fixed from the humerus growth cartilage of newborn piglets.

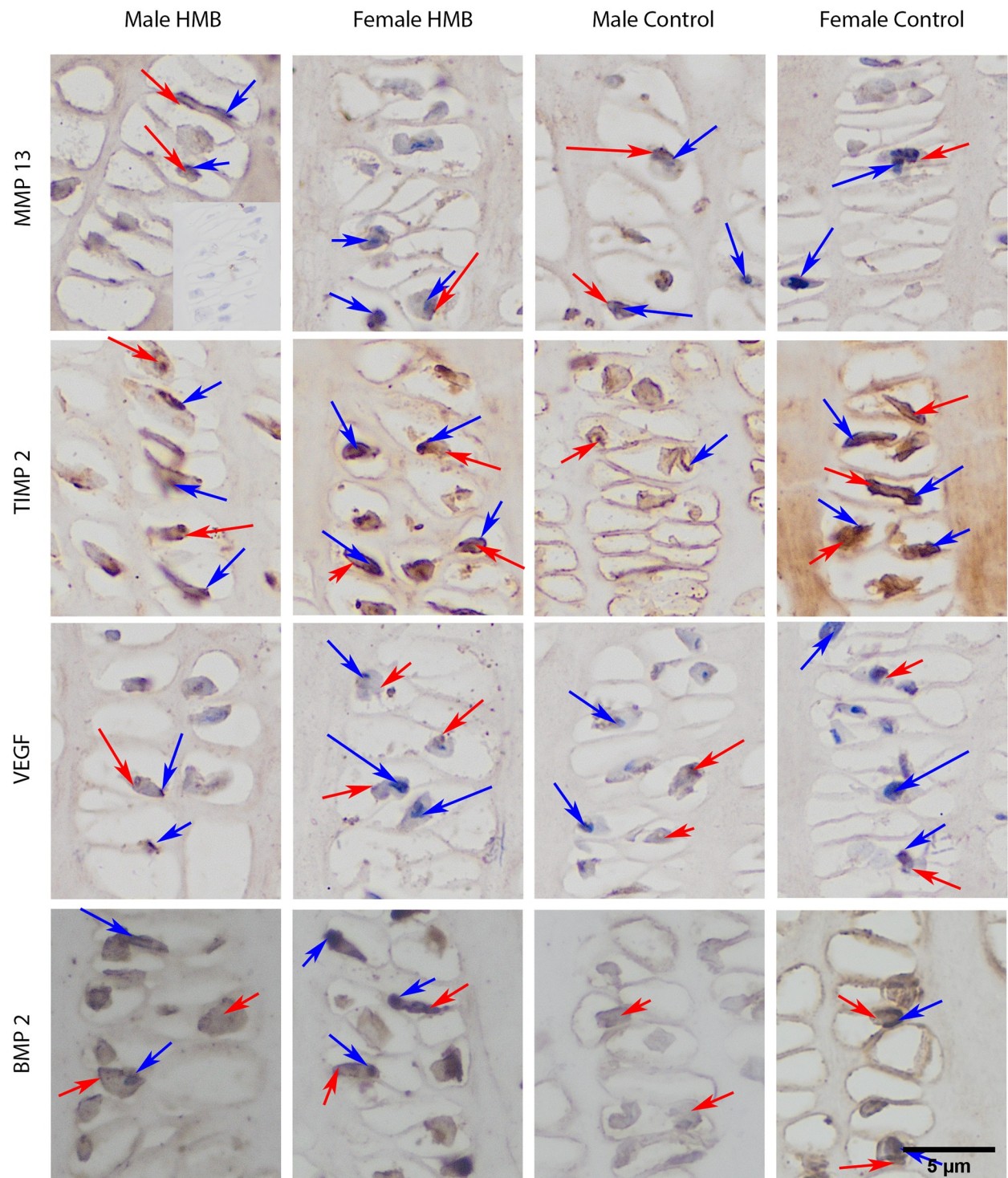

**Fig 4. Immunohistochemical analysis of the immunolocalization the MMP13 (matrix metalloproteinase 13), the TIMP2 (tissue inhibitor of metalloproteinases 2), the VEFG (vascular endothelial growth factor) and the BMP2 (bone morphogenetic protein 2) from the proliferative zone of growth plate cartilage.** Representative images immunohistochemical analysis carried out on the formaldehyde-fixed from the humerus growth cartilage of newborn piglets. The positive reaction (presence of MMP13, TIMP2, VEFG, BMP2) in the cytoplasm or cell nucleus was marked in red, while the lack of expression was marked in blue.

groups, TIMP2 expression is more pronounced in the group whose mothers did not receive HMB (Table 7, Figs 3 and 4).

A stronger response in the matrix of growth plate cartilage in zones III and IV was observed in all groups. Analysis of TIMP expression in the growth plate matrix showed a decrease in TIMP expression in both HMB groups compared to the control (Table 8, Figs 3 and 4).

## Bone morphogenetic protein 2 (BMP2) immunolocalization in the bone trabecula and the growth plate cartilage

The BMP2 staining showed expression of this protein in the matrix of and bone trabecular cells in all groups tested, compared to the negative control. Statistical analysis in the case of a positive response in the cytoplasm of cells showed no differences between the studied groups. However, microscopic observation showed that maternal supplementation increased BMP2 expression in the cytoplasm of trabecular bone cells in their offspring, regardless of gender (Table 7, Fig 2).

Analysis of the expression of this protein in the bone trabecula matrix showed no differences between the studied groups (Table 8, Fig 2).

In the case of the cartilage, staining for BMP2 showed its presence in all animals both in the matrix and in the cytoplasm of cells as well as its absence in the nuclei of the cell Statistical analysis in the case of a positive response to the presence of BMP2 in the cytoplasm of cells showed that males and females from the HMB groups had a greater response compared to the control groups. Comparing both sexes not dependent on supplementation, males were characterized by a stronger expression of the tested protein in the cytoplasm of cells (Table 7, Figs 3 and 4). Gray-scale analysis showed higher BMP2 expression in the growth plate matrix in the HMB groups (Table 8, Figs 3 and 4).

## Discussion

The currently available literature also shows that changes in mother's nutrition significantly affect the process of the formation of the skeleton system of their offspring. Human studies show that inadequate maternal nutrition during pregnancy negatively affects the formation of peak bone mass during prenatal fetal development [2, 39].

The nutritional factor used in this experiment is HMB supplementation. The use of this supplement during pregnancy contributes to a significant increase in the body weight of newborn organisms and a proportional increase in the mass of internal organs. The same research shows that maternal supplementation affects the development of the femur in newborn animals [17]. In this experiment, the sows were supplemented between 70 and 90 days of pregnancy with HMB at a daily dose of 0.2 g/kg b.w. However, there is no information as to whether this supplement will affect the humerus. Although studies conducted in animal models show that the growth of the humerus is the fastest and slows down much earlier compared to other long bones [1]. Therefore in the presented experiment was justified to use the same dose of this supplement in the same period of pregnancy.

There are a few papers about the HMB influence with animal model usage wherein the authors decided to use of pig model, which is considered to be the best animal model used in particular in preclinical and clinical feeding. These animals are characterized by the greatest similarity to humans in terms anatomical and physiological. In particular, the structure of the digestive tract, cardiovascular, respiratory, immune and urinary systems. Also their internal organs are of similar size and perform similar functions [3].

In the presented study, an increase in humerus weight in newborns caused by the supplementation with HMB of their mothers during pregnancy by was observed by 84% in males

and by 51% in females, respectively. Regarding the length of the examined humerus, we observed it is increase only in males by 22%. Only under the effect of HMB the increase in the value of the bone geometric properties was observed. In males, geometric parameters such as vertical and horizontal external diameters, cross-sectional area, the mean relative wall thickness, the cortical index and the moment of inertia increased by 22%, 18%, 52%, 36%, 12%, 89% respectively. However, in the case of females from the experimental group in relation to the control group, only the geometrical property of the examined bone that has changed is the moment of inertia, which increased by 45%. Changes in bone geometrical properties are associated with changes in their morphology, they are higher in males, which could have contributed to greater changes in these parameters in this sex. In femoral studies of newborn animals whose mothers were supplemented with HMB at the same dose and over the same period of time, the increase in bone mass was proportional for both sexes. In the case of the humerus, the effect of supplementation is not proportional and is higher in males compared to females by 19%. Differences were also observed in geometric parameters such as the cross-sectional area, the mean wall thickness and the cortical index, in which the effect of supplementation was lower in the humerus than in the femur. For both long bones tested, males showed a greater effect on maternal supplementation on geometrical properties. The observed differences between the humerus and femur in HMB groups may be associated with different prenatal development of these bones [1, 17].

Changes in the humeral bone morphology and geometric properties in newborn males from the experimental group improved the mechanical properties of both ultimate strength and maximum elastic strength by 71% and 113%, respectively. In contrast, females from the HMB group had a lower increase in ultimate strength and maximum elastic strength by 49% and 85%, respectively. Both of these forces are significantly higher in males, which is associated with greater changes in the weight, length and geometrical properties of the humerus in this sex. An increase in mechanical properties was also observed in the femur, where in male ultimate strength and maximum elastic strength increased by 70% and 77,6%, respectively, while in females ultimate strength increased by 33,7%. However, in the case of the humerus we are observed larger changes in mechanical properties compared to the femur [17].

In the presented work, we also examined the effect of mother's supplementation during pregnancy on the collagen structure of the cortical bone. Changes in collagen structure can contribute to changes in their ability to resist the force affecting the skeleton. In males, under the influence of the supplementation, a decrease in the share of mature collagen fibers and a ratio of mature to immature collagen was observed by 43% and 55%, respectively. In contrast, in the HMB females group only an 42% increase of immature collagen fibers was observed. The results showed that sex significantly influenced the effect of supplementation because such differences were not observed between the sexes in the control group. Changes in the collagen structure of the cortical bone may have contributed to the appearance of changes in the examined mechanical properties of the humerus. Immature collagen fibers undergo the maturation process and is less mineralized than the mature collagen fibers [34]. We observed the increase in the share of immature collagen fibers in the examined females and a decrease in the content of mature collagen fibers in males. This could have contributed to changes in individual mechanical parameters of the humerus under the influence of HMB supplementation.

Histomorphometric analysis of trabecular bone showed that maternal supplementation only significantly increased BV/TV in females by 63%, and in males reduced bone thickness by 9% at the humerus epiphysis. However, no changes in trabecular bone histomorphometry were observed in the humerus metaphysis. In prenatal femoral studies, maternal supplementation during pregnancy contributed to major changes in trabecular bone histomorphometry at both the metaphysis and the epiphysis. Where in the femoral epiphysis an increase in the

number of bone trabeculae was observed only in males. Other parameters such as BV/TV, thickness of bone trabeculae in both sexes increased significantly while the size of the trabecular space decreased. On the other hand, in the femoral metaphysis, changes in trabecular bone histomorphometry was observed only in female [17]. The observed differences between the two long bones may be due to different development during fetal life [1]. It should be noted that the changes in the trabecular histomorphometry of the humerus are much smaller than in the case of the femur. This may be due to the fact that the bone develops earlier than the femur and was less intense at the time supplementation began.

On the other hand, we observed changes in the collagen structure of the trabecula bone the epiphysis and the metahysis. In the HMB males group at the epiphysis the share of immature and mature collagen fibers decreased, which contributed to a decrease in their ratio compared to the control group by 45%, 54% and 44%, respectively. In the HMB females group, only a 37% increase of the immature collagen fibers was observed. In the metaphysis of the HMB males, the share of the immature collagen fibers was reduced by 62%, while there was no change in the share of the mature collagen fibers, which contributed to an increase in the ratio of these collagens by 33%. In females in the metaphysis, maternal supplementation increased the proportion of the immature and mature collagen fibers by 50% and 40%, respectively. These changes show an increase in the intensity of bone tissue remodeling processes due to the supplementation of their mothers during pregnancy.

Under the influence of the female supplementation during pregnancy, changes in the articular and the growth plate cartilage were also observed. Concerning the articular cartilage in males from the experimental group, an increase in the thickness of all zones was observed, i.e., I, II and III, respectively, by 41%, 13%, 54% in relation to the control group, which was subjected to an increase in the total thickness of articular cartilage in this experimental group by 9%. In the case of females from the HMB group, we observed an increase in the thickness of the zone I, II and the total thickness of the articular cartilage by 19%, 18%, and 11% in relation to the females from the control group. HMB supplementation also affected BTD, but only in males, which was significantly lower by 21% compared to the control group. These changes may affect functions that individual zones of this cartilage perform [40, 41]. Changes in the thickness of individual zones can affect the functions of articular cartilage, but it should be noted that the increase in thickness of individual zones is not proportional when comparing each of them with each other in both sex.

In the case of the growth plate cartilage, we also observed changes in both experimental groups. In females of this group we observed an increase in the thickness of zones I, II, III in a total of 15%, 27%, 36%, 16%, respectively. In males, the thickening under the influence of the HMB was seen in zones II, III by 23% and 31%, with a decrease in the thickness of the zone I by 25%, and a lack of changes in relation to the control group in the total thickness of the articular cartilage. In the case of these measurements, we see that the HMB had a greater impact in females. In the articular cartilage the individual zone size changes may affect the proper functioning of the entire growth plate cartilage, which is the growth plate, and contributes to the bone growth as a result of the cartilage ossification [42].

Changes that we observed may be the cause of the effect of the tested supplement on the body's hormonal balance, as confirmed by the results obtained by Blicharski et al. [17]. In this work the HMB affects the concentration of IGF-1, leptin and sex hormones, contributing to the increase in the content of these hormones in the blood serum.

Maintaining the proper relationship between osteolytic and osteogenetic processes significantly affects the correct formation of the skeletal system. Normal bone and cartilage resorption is affected by maintaining a balance between MMP13 and TIMP2. Proteins from the MMP group are responsible for resorption processes of tissue matrix components. In contrast,

proteins from the TIMP group that inhibit the effects of MMPs. The MMP group includes 14 proteins that have been divided according to their structure and function. Lack of certain proteins from this group, including MMP13 (collagenase 3) during embryonic development, has a negative effect on the formation of growth plates, which impairs bone development and growth [43]. The TIMP family includes 4 proteins that differ in terms of expression, regulation as well as the ability to interact with a given metalloproteinase found in a tissue matrix in a latent form [44]. Therefore, in this study it was examined whether the changed nutrition of mothers during pregnancy affects the expression of metalloproteinases, especially MMP13 in bone and cartilage tissue of newborn animals as well as its TIMP2 inhibitor.

In the presented work, HMB supplementation of mothers during pregnancy significantly influenced the expression of both MMP13 and TIMP2. In the trabecular bone, changes in MMP13 expression are observed only in females, where supplementation contributed to increased expression of this protein. However, in the case of the administration of the growth plate in the mother during pregnancy HMB increased expression of this protein in the cytoplasm of chondrocytes their offspring irrespective of sex. In all examined groups, MMP13 expression in the growth plate matrix was the strongest on the border of zones III and IV.

Studies in animal models show that mice in which the gene encoding the MMP13 has been deleted created disturbances in the bone and the cartilage tissue homeostasis with no change in their life expectancy and fertility compared to the control group. In individuals deprived of MMP13, disorders in the growth plate were found as a result of changes in the thickness of the hypertrophic zone and no change in the thickness of the other zones, which caused an increase the number of trabecular bone in the absence of regularity in their arrangement, and disorders in their proper ossification, contributing to weakening their strength and increasing risk of fracture over the length. At the same time, these changes did not affect bone length [26, 45].

MMP13 affects the degradation of collagen fibers, mainly type II but also type I, which build support tissues. Changes in the expression of this protein as a result of maternal supplementation can also be combined with analysis of trabecular bone collagen structure, where changes were observed under the influence of HMB supplementation [26, 45, 46].

In the results obtained in this study, it was observed that in the control group, TIMP2 expression is significantly stronger than the immunological reaction, we obtained in the experimental group in both cartilage and trabecular tissue, which may indicate an increase in resorption processes in these tissues under the influence of HMB. This can also be confirmed by the fact that MMP13 expression is stronger in the HMB supplemented group in both cartilage and trabecular bone. Research conducted in recent years shows that reduced TIMP levels contribute to increased bone resorption through stronger expression of metalloproteinases [46].

Research conducted over the years shows that the expression of MMP13 is not only correlated with the expression of proteins from the TIMP group but also among others with BMP2 [47]. BMP2, the second type of bone morphogenetic protein, plays an important role in the formation of both the bone and the cartilage tissue. It is a protein that stimulates the formation of new bone tissue during bone remodeling [24, 25]. It is released during collagen degradation and stimulates its reconstruction. Research conducted by Johansson et al. [1997] shows that the BMP2 protein not only stimulates the formation of type II collagen, but also significantly inhibits the action of collagenase 3 [47].

In this work, we can see that the BMP2 protein expression is stronger in the experimental group compared to the control group. It should be noted that males, regardless of supplementation, were characterized by a stronger expression of this protein in both cartilage cells and matrix, especially, stronger BMP2 expression in between zone III and IV was observed. The

obtained results may suggest that bone formation processes occur much faster in animals whose mothers were supplemented with the HMB during pregnancy.

Increased expression of both BMP2, which inhibits the action of MMP13 and contributes to the increased synthesis of collagen fibers, as well as collagenase 3 degrading collagen fibers in bone and cartilage tissue in the offspring of mothers supplemented with HMB during pregnancy may suggest that osteolytic as well as osteogenetic processes occur much faster in these newborns as a result of maternal supplementation. It should also be noted that the expression of the TIMP2 metalloproteinase inhibitor is weaker in the experimental groups, which may also confirm the fact of increased remodeling processes of supporting tissues in these animals.

Studies using BMP2 have shown that its activity is highest during endochondral bone development, because it affects the proliferation and maturation of cartilage cells. Studies in animal models show that the lack of BMP2 negatively affects the development of the skeletal system, which is associated with disorders in the growth plate. The hypertrophic zone in the course of growth is very important because in this part the blood vessels enter into the growth plate, stimulating the growth of bone cells, their ossification and the formation of a new matrix [48]. The appearance of blood vessels stimulates bone cells and other cells for production the right amount of growth factor, which is VEGF [49]. The VEGF, or the endothelial growth factor, plays crucial role in the endochondral bone development. Research conducted by Carlevaro et al in 2000 proved that the highest VEGF expression is observed in the hypertrophic zone and stimulates bone growth in length [49–51].

In this studies all animals, irrespective of group and sex, VEGF was found in the matrix and nuclei of cartilage cells in the II and III growth cartilage zones. The highest expression of this protein was observed at the border of zones III and IV, which may confirm that this factor stimulates intra-cartilaginous growth. The strongest expression of this protein at the border of the hypertrophic and the calcifying zone can be observed in the HMB females, which may indicate the effect of both supplementation and sex on the VEGF expression, because the same effect is not observed in males. On the other hands larger changes in osteometric parameters under the influence of the HMB are visible in males. Result could be associated with the greater BMP2 expression in this sex.

There is no research on the effect of HMB on the expression of MMP13, TIMP2, BMP2 and VEGF in the literature.

It should be noted that proteins identified in this study were characterized by strong expression in zones II and III and on the border of zones III and IV of growth plate cartilage, which can also confirm that these proteins affect the process of bone growth in length. Obtained results that changes in maternal nutrition affect the expression of MMP13, TIMP2, BMP2 and VEGF in the bone and cartilage tissue of their newborn offspring, which may contribute to changes in the development of their skeletal system [2].

Maintaining balance in osteolytic and osteogenic processes makes the proper bone tissue homeostasis and proper growth of this bone tissue [52]. The obtained results suggest that the bone remodeling processes are significantly more intensive in the experimental group compared to the control group. What can be combined with the previously studies where mass and length of the femur bone, are significantly higher in animals from the experimental group [17]. We also observed changes in the thickness of both the articular and the growth cartilage, which can be associated with changes in the expression of proteins MMP13, TIMP2, BMP2 and VEGF.

To summarize there is no research in the available literature answering the question whether HMB supplementation of females during pregnancy affects the development of prenatal humerus bone. The comparison of the results obtained in this work in relation to the changes caused by the HMB supplementation on the prenatal development of the femur gives

the view that the humerus bone responds in a different way to changes in mother's nutrition, which may be due to the fact that the prenatal development of both bones is different. It should also be noted that checked whether the effect of femoral changes under the influence of HMB persists later in life as well [16]. Therefore, further researches in this area should also be carried out to verify if these changes are equally observable in the humerus. Another aspect of our work that was not previously used in assessing the impact of maternal supplementation on prenatal development of the skeletal system of their offspring, was the study of the protein expression (TIMP2 MMP13, VEGF, BMP2). These proteins, in studies assessing the impact of adequate nutrition of mothers on the development of their offspring, can be an excellent markes of the intensity of osteogenic and osteolytic in both the cartilage and the bone tissue. Our research may be an introduction to the further research on the similarity between the femur and the humerus in regard to their changes during prenatal development and adaptations to changed conditions.

## Author Contributions

**Conceptualization:** Agnieszka Tomczyk-Warunek, Tomasz Blicharski, Jaromir Jarecki, Siemowit Muszyński, Ewa Tomaszewska, Lucio C. Rovati.

**Data curation:** Agnieszka Tomczyk-Warunek, Tomasz Blicharski, Piotr Dobrowolski.

**Formal analysis:** Tomasz Blicharski, Piotr Dobrowolski, Siemowit Muszyński, Ewa Tomaszewska, Lucio C. Rovati.

**Funding acquisition:** Tomasz Blicharski, Siemowit Muszyński.

**Investigation:** Agnieszka Tomczyk-Warunek, Tomasz Blicharski, Jaromir Jarecki, Piotr Dobrowolski, Ewa Tomaszewska.

**Methodology:** Agnieszka Tomczyk-Warunek, Tomasz Blicharski, Piotr Dobrowolski, Ewa Tomaszewska.

**Project administration:** Tomasz Blicharski, Piotr Dobrowolski, Siemowit Muszyński, Ewa Tomaszewska.

**Resources:** Tomasz Blicharski.

**Supervision:** Tomasz Blicharski, Ewa Tomaszewska.

**Validation:** Agnieszka Tomczyk-Warunek, Tomasz Blicharski, Jaromir Jarecki, Siemowit Muszyński, Ewa Tomaszewska.

**Writing – original draft:** Agnieszka Tomczyk-Warunek, Tomasz Blicharski, Jaromir Jarecki, Piotr Dobrowolski, Siemowit Muszyński.

**Writing – review & editing:** Agnieszka Tomczyk-Warunek, Tomasz Blicharski, Siemowit Muszyński.

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
