## [Decision Letter · Decision Letter 0]

1 Dec 2020

PONE-D-20-30549

The effect of maternal HMB supplementation on bone mechanical and geometrical  properties, as well as histomorphometry and immunolocalization of VEGF, TIMP2, MMP13, BMP2 in the bone and cartilage tissue of the humerus of their newborn piglets.

PLOS ONE

Dear Dr. Blicharske,

Thank you for submitting your manuscript to PLOS ONE. After careful consideration, we feel that it has merit but does not fully meet PLOS ONE’s publication criteria as it currently stands. Therefore, we invite you to submit a revised version of the manuscript that addresses the points raised during the review process.

Both reviewers identified novelty and methodology issues in the current form of manuscript. Please carefully read their comments and provide point-by-point responses.

We look forward to receiving your revised manuscript.

Kind regards,

Xiaofang Wang

Academic Editor

PLOS ONE

Journal Requirements:

3. Please ensure you have thoroughly discussed any potential limitations of this study within the Discussion section, including the potential impact of confounding factors.

4. At this time, we request that you please report additional details in your Methods section regarding animal care, as per our editorial guidelines: 1) please describe any steps taken to minimize animal suffering and distress, such as by administering analgesics. Thank you for your attention to these requests.

"NO"

6. Please upload copies of Figures 3 and 4, to which you refer in your text (line 364 and line 370). If the figures are no longer to be included as part of the submission please remove all reference to them within the text.

Reviewers' comments:

Reviewer's Responses to Questions

**Comments to the Author**

1. Is the manuscript technically sound, and do the data support the conclusions?

Reviewer #1: Yes

Reviewer #2: Yes

2. Has the statistical analysis been performed appropriately and rigorously? 

Reviewer #1: Yes

Reviewer #2: Yes

3. Have the authors made all data underlying the findings in their manuscript fully available?

Reviewer #1: Yes

Reviewer #2: Yes

4. Is the manuscript presented in an intelligible fashion and written in standard English?

Reviewer #1: Yes

Reviewer #2: Yes

5. Review Comments to the Author

Reviewer #1: The authors reported the impacts of HMB ( hydroxy-β-methobutyrate ) supplementation of mothers during pregnancy on the development of the skeletal system of their offspring using sows of the Great White Poland breed. As they mentioned there were some similar studies reported recently using the model while they claimed they focused on humerus bones instead of femur bones. So the novelty is not high for current study. Moreover, it will be interesting to demonstrate the different impacts of HMB on humerus and femur bones if possible. Moreover, the authors tested the expression of some proteins, including VEGF, TIMP2,MMP13, BMP2 using the histomorphometry and immunolocalization methods. As we know, these staining-based methods are not good for quantitively analysis. It is challenging to compare their expression levels based on their current data.

Reviewer #2: General comments

The authors studied the effect of hydroxy-β-methobutyrate maternal supplementation on the immediate postnatal humeral bone, growth plate cartilage and articular cartilage phenotypes of the offspring in a pig model. Supplementation led to greater bone mass and length as well as increases in other architectural variables, collagen structure and protein expression. Some of the positive effects of the supplementation were sex-dependent. This is a careful descriptive study.

Specific comments

1. The ethics statement is incomplete: “If anesthesia, euthanasia, or any kind of animal sacrifice is part of the study, include briefly which substances and/or methods were applied.”

2. Tables showed pooled SEM’s, but it would be much better to include the standard deviation for each individual mean value.

3. Some references to figures in the text did not include the figure number.

4. The Discussion is very long, and, perhaps, could be shortened by not repeating the findings while still providing the comparisons to the literature and interpretations.

6. PLOS authors have the option to publish the peer review history of their article (what does this mean?). If published, this will include your full peer review and any attached files.

Reviewer #1: No

Reviewer #2: No

---

## [Author Response · Author response to Decision Letter 0]

19 Jan 2021

Reviewer 1

Thank you very much for your review. All comments contributed to the introduction of valuable changes to the manuscript. The work was supplemented with two additional analyzes that are used to evaluate the expression of proteins in tissues in immunohistochemical methods. The first of these methods was the method consisting in assessing the participation of cells with positive responses to the presence of VEGF, TIMP2, MMP13, BMP2. This analysis consisted in designating 10 fields of observation in which all cells were counted. Then, cells with a positive reaction to the presence of tested proteins were counted and their percentage in relation to all cells was calculated. The second method used to study the expression of VEGF, TIMP2, MMP13, BMP2 in the matrix of growth plate cartilage and bone trabeculae was the analysis of gray, which is based on the brightness of pixels. The higher the pixel value, the weaker the expression of the tested protein. Both methods confirmed the previously performed observation that HMB supplementation influences to the expression of VEGF, TIMP2, MMP13, BMP2 both in the growth plate cartilage and the humerus bone trabeculae. Both methods used to analyze the expression of the studied proteins, both in cells and in the matrix of bone and cartilage tissue, are standard methods used in studies of the expression of these proteins. Both methods are standard methods used in studies of the expression of proteins.

In the presented manuscript, the results of the research obtained in our study were compared with the results published by Blicharski et al. in 2017. In this publication the effects of maternal supplementation on the prenatal development of the femur of their offspring were tested on the same model. The comparison of the results shows that the humerus responds differently to the changes in maternal nutrition, which may be due to the fact that the prenatal development of this two long bones is other. The obtained results are new as there is no literature information regarding the influence of maternal HMB supplements on the prenatal development of the humerus and they complement the available knowledge. 

Reviewer 2

Thank you for your review. All comments have been incorporated in the revised manuscript.

Detailed comments:

1. The method of euthanizing animals has been completed

2. All tables were complete for standard deviation

3. The numbering of figures and gaps in the text were checked

4. The discussion was shortened by 1,5 pages

---

## [Decision Letter · Decision Letter 1]

8 Feb 2021

The effect of maternal HMB supplementation on bone mechanical and geometrical  properties, as well as histomorphometry and immunolocalization of VEGF, TIMP2, MMP13, BMP2 in the bone and cartilage tissue of the humerus of their newborn piglets.

PONE-D-20-30549R1

Dear Dr. Blicharski,

We’re pleased to inform you that your manuscript has been judged scientifically suitable for publication and will be formally accepted for publication once it meets all outstanding technical requirements.

Kind regards,

Xiaofang Wang

Academic Editor

PLOS ONE

Additional Editor Comments (optional):

Reviewers' comments:

Reviewer's Responses to Questions

**Comments to the Author**

1. If the authors have adequately addressed your comments raised in a previous round of review and you feel that this manuscript is now acceptable for publication, you may indicate that here to bypass the “Comments to the Author” section, enter your conflict of interest statement in the “Confidential to Editor” section, and submit your "Accept" recommendation.

Reviewer #1: All comments have been addressed

Reviewer #2: All comments have been addressed

2. Is the manuscript technically sound, and do the data support the conclusions?

Reviewer #1: Yes

Reviewer #2: Yes

3. Has the statistical analysis been performed appropriately and rigorously? 

Reviewer #1: Yes

Reviewer #2: Yes

4. Have the authors made all data underlying the findings in their manuscript fully available?

Reviewer #1: Yes

Reviewer #2: Yes

5. Is the manuscript presented in an intelligible fashion and written in standard English?

Reviewer #1: Yes

Reviewer #2: Yes

6. Review Comments to the Author

Reviewer #1: (No Response)

Reviewer #2: The Discussion is still very long and much of it simply repeats the Results. There may be a way to shorten this, but that is a decision for the Editor.

7. PLOS authors have the option to publish the peer review history of their article (what does this mean?). If published, this will include your full peer review and any attached files.

Reviewer #1: No

Reviewer #2: No

---

## [Editor Report · Acceptance letter]

15 Feb 2021

PONE-D-20-30549R1 

The effect of maternal HMB supplementation on bone mechanical and geometrical  properties, as well as histomorphometry and immunolocalization of VEGF, TIMP2, MMP13, BMP2 in the bone and cartilage tissue of the humerus of their newborn piglets 

Dear Dr. Blicharski:

I'm pleased to inform you that your manuscript has been deemed suitable for publication in PLOS ONE. Congratulations! Your manuscript is now with our production department. 

Kind regards, 

on behalf of

Professor Xiaofang Wang 

Academic Editor

PLOS ONE